# Group Vision Transformer

## ABSTRACT

The Vision Transformer has attained remarkable success in various computer vision applications. However, the large computational costs and complex design limit its ability in handling large feature maps. Existing research predominantly focuses on constraining attention to small local regions, which reduces the number of tokens attending the attention computation while overlooking computational demands caused by the feed-forward layer in the Vision Transformer block. In this paper, we introduce Group Vision Transformer (GVT), a relatively simple and efficient variant of Vision Transformer, aiming to improve attention computation. The core idea of our model is to divide and group the entire Transformer layer, instead of only the attention part, into multiple independent branches. This approach offers two advantages: (1) It helps reduce parameters and computational complexity; (2) it enhances the diversity of the learned features. We conduct comprehensive analysis of the impact of different numbers of groups on model performance, as well as their influence on parameters and computational complexity. Our proposed GVT demonstrates competitive performances in several common vision tasks. For example, our GVT-Tiny model achieves 84.8% top-1 accuracy on ImageNet-1K, 51.4% box mAP and 45.2% mask mAP on MS COCO object detection and instance segmentation, and 50.1% mIoU on ADE20K semantic segmentation, outperforming the CAFormer-S36 model by 0.3% in ImageNet-1K top-1 accuracy, 1.2% in box mAP, 1.0% in mask mAP on MS COCO object detection and instance segmentation, and 1.2% in mIoU on ADE20K semantic segmentation, with similar model parameters and computational complexity. Code is accessible at github.com/AnonymousAccount6688/GVT.

## KEYWORDS

Vision Transformer, Image Classification, Group FFN

## 1 INTRODUCTION

Convolutional neural networks (CNNs) have become prevailing architectures for computer vision applications, encompassing tasks such as image classification, object detection, and semantic segmentation. Following the remarkable success of AlexNet [17] in ImageNet-1K classification, there has been a continuous evolution toward developing deeper and more efficient CNNs [12, 27, 28] to enhance image recognition capabilities. In pursuing greater efficiency within CNN models, a succession of methodologies [14, 16, 29, 35] has been developed to mitigate model parameters and floating-point operations (FLOPs).

Meanwhile, inspired by the remarkable success of Transformer [31] in the field of natural language processing (NLP), a multitude of studies have embarked on incorporating self-attention mechanisms into computer vision tasks. The Vision Transformer has emerged as a predominant model in computer vision applications. ViT [9] initially partitions an input image into multiple patches, transforming each patch into a token; subsequently, a Transformer encoder is employed to process these tokens utilizing the self-attention (SA) mechanism, enabling capturing both global and local relationships within the image.

The Vision Transformer is composed of multiple stacks of basic blocks, each consisting of self-attention and a multilayer perceptron (MLP). The computational mechanism of self-attention shows linear complexity concerning the dimension of the token but exhibits quadratic complexity with respect to the number of tokens. Consequently, a series of methodologies sought to mitigate the computational complexity associated with self-attention. Another disadvantage of the Vision Transformer, when compared to Convolutional Neural Networks (CNNs), is the absence of inductive bias, as it lacks spatial hierarchies and translational invariance. Note that the Vision Transformer attempts to implicitly learn inductive bias from a large amount of training data, such as JFT-300. Recently, local attention Transformers have emerged to establish local inductive bias while concurrently reducing computational demands.

In the Swin Transformer [21], a non-overlapping window attention mechanism was introduced to concentrate on a pre-defined window area. Additionally, a shifted window mechanism was proposed for token mixing between two adjacent windows. Wang et al. [32] utilized a progressive shrinking pyramid to alleviate computation associated with large feature maps. Dong et al. [8] introduced a cross-shaped attention that performs self-attention in both horizontal and vertical stripes by dividing multi-heads into parallel groups. Ramachandran et al. [25] applied self-attention to replace traditional convolutional operations, where each pixel attends to the attention computation within a window, utilizing a sliding window. Furthermore, Hassani et al. [10] presented an efficient and memory-friendly local attention mechanism to reduce attention computation while preserving local inductive bias.

Despite the ability of the aforementioned approaches to reduce computation by concentrating on small regions, they tend to overlook the channel dimension. In the Vision Transformer, each Transformer block comprises two main parts: the attention layer and the feed-forward network (FFN). The computational complexity of the attention part is quadratic concerning the token length and linear with respect to the channel dimension, while the computational complexity of the FFN part is quadratic concerning the channel length. The model complexity induced by the FFN layer is even larger than the self-attention layer. For instance, the NAT tiny model [10] comprises 28M parameters and 4.3G FLOPs, of which 12.461M parameters and 1.796G FLOPs are attributed to the self-attention layer, with the remaining generated by the feed-forward network. In a recent work [20], a cascade group attention was proposed to reduce computation and diversify features by segmenting the feature map into multiple heads; attention is then computed for each head without sharing parameters. However, the emphasis of these studies remains predominantly on attention computation, with insufficient attention directed to the feed-forward network.

In this paper, we propose a new Group Vision Transformer (GVT). We first partition the input feature tokens into multiple groups

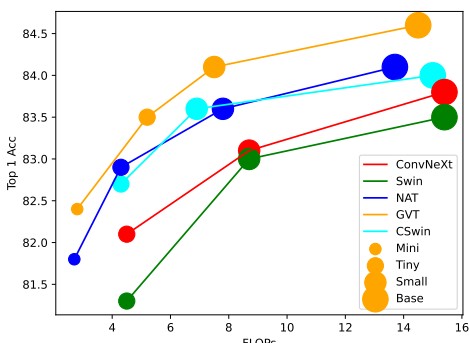

**Figure 1: Comparison of Top-1 Accuracy on ImageNet-1K classification and model complexity. Our GVT outperforms NAT and Swin with similar parameters and computational complexity.**

along the token dimension. Next, we apply self-attention and feed-forward network (FFN) operations independently on each feature group. Simultaneously, one group is dedicated to capturing the global dependencies of the shallow layer. This design offers several advantages. First, the grouped feature tokens result in a reduction of computation in the FFN, which accounts for the majority of the Transformer Block computation, by a factor of $g$, where $g$ denotes the number of groups. Second, each feature group attends to self-attention and FFN computations independently, thereby enhancing the diversity of the feature groups. To incorporate inductive bias and global-range dependencies in the shallower layers, one group is utilized to capture global contours while the remaining groups are responsible for capturing local details. Experimental results across various computer vision tasks, including image classification, object detection, instance segmentation, and semantic segmentation, demonstrate that the proposed Group Vision Transformer achieves competitive performance compared to baselines while maintaining comparable model parameters and computational complexity (e.g., Fig. 1).

## 2 RELATED WORK

This section gives a brief review of Vision Transformers, highlights notable instances of Group Convolutions, and discusses recent advances of efficient Vision Transformers.

### 2.1 Vision Transformer

The Vision Transformer was first proposed in [9], where an input image is divided into small non-overlapping patches. These patches are then linearly embedded and treated as sequences of tokens. The tokenized patches undergo processing through multiple layers of Vision Transformer (ViT) blocks to capture spatial relationships among the feature tokens and learn relevant features. A ViT block, a fundamental building unit in the Vision Transformer architecture, comprises two components: A self-attention layer and a feed-forward network (FFN). In the self-attention layer, attention scores for each token of the image are determined based on its

relevance to all the other tokens. Given an input token $X \in R^{n \times d}$, where $n$ is the length of the token and $d$ is the dimension of each token, three matrices $Q, K$, and $V$ are defined to map the input token to a query and a key-value pair. The output is computed as the weighted sum of the value matrices [31]. The output matrix is calculated as:

$$Attention(Q, K, V) = Softmax(\frac{QK^T}{\sqrt{d}})V, \qquad (1)$$

whose computation has a complexity of $O(n^2 d)$.

After the self-attention layer, a two-layer Position-wise feed-forward network is added to introduce more non-linearity to the model and enhance the representation ability of the model, as well as for dimension interaction. This allows the model to learn complex, non-linear relationships within the feature token. Given an input token $X \in R^{n \times d}$, the FFN is computed as:

$$FFN(X) = \delta(XW_1 + b_1)W_2 + b_2, \qquad (2)$$

where $W_1$ and $W_2$ denote the parameters of the two MLP layers in the FFN, $b_1$ and $b_2$ represent the biases of the two MLP layers respectively, and $\delta$ denotes a non-linear activation operation (e.g., GELU [13]). The FFN has a computational complexity of $O(nd^2)$.

### 2.2 Group Convolution

Group convolution was initially introduced in AlexNet [17], where it was employed to distribute the model across two GPUs. In [26], depth-wise separable convolution was proposed to reduce parameters and computation by first conducting convolutions on each feature map independently and then mixing the features across different feature maps using a $1 \times 1$ convolution [18]. Xie et al. [35] introduced a homogeneous multi-branch structure utilizing group convolutions to enhance performance while preserving model parameters and computational complexity. Afterwards, group convolution was applied to develop more efficient models for edge devices, such as MobileNet [14] and Xception [4], as replacements for traditional convolutions. In [39], a shuffle unit was proposed to enhance information flow among different groups, thereby improving the overall performance.

### 2.3 Efficient Vision Transformers

The original Vision Transformer (ViT) model [9] computes the attention scores of a token with all the other tokens, resulting in quadratic computational complexity concerning both the token length and embedding dimension. Afterwards, researchers have sought to reduce the computation of the Vision Transformer and develop more efficient ViT models.

In the Swin Transformer [21], attention computation is focused only on a small window rather than the entire pixel space. In PVT [32], spatial reduction is employed on the key and value matrices, thereby reducing the computation required for attention. In SASA [25], a sliding local attention module was proposed, replacing all convolutions in ResNet [12] with a self-attention layer, resulting in improved performance with less parameters and computation. In [8], a cross-shaped window self-attention was proposed, where token self-attention calculations are conducted in horizontal and vertical stripes in parallel. Each stripe is obtained by splitting the

input features into stripes of equal width, leading to improved performance with the similar parameters and computational demands. Hassani et al. [10] improved SASA [25] by providing a memory- and speed-efficient implementation.

## 3 METHOD

This section presents our Group Vision Transformer (GVT).

### 3.1 The Overall Architecture

The overall architecture of our Group Vision Transformer is shown in Fig. 2. Following the common practice in NAT [10], we split an input image into overlapped patches by a patch splitting module. Each patch is regarded as a token and is derived as the concatenation of the pixel RGB values. This is implemented by two layers of strided convolution, with a stride of 2 and a kernel size of 3. The patch size is set to 4. As a result, the feature dimension of each token is $4 \times 4 \times 8 = 128$. To attain a hierarchical representation, the whole network consists of four stages. In each stage, a token will be processed by a convolution to reduce the spatial resolution and double the feature dimension using a strided convolution (kernel size $3 \times 3$, stride $2 \times 2$). Therefore, the $i$-th stage will generate a feature map of size $\frac{H}{2^i} \times \frac{W}{2^i} \times C \cdot 2^i$, where $H$, $W$, and $C$ represent the height, width, and channels of the feature map generated by the first stage. Each stage consists of $N_i$ sequential *Group Vision Transformer Blocks*, and the number of tokens and feature dimension are maintained within each stage. Consequently, the proposed architecture can be applied to various downstream vision tasks, such as object detection, instance segmentation, and semantic segmentation.

### 3.2 Group Self Attention

Despite the fact that many previous ViT studies sought to reduce the computational complexity by focusing the attention computation on small local windows, they neglected to consider the token dimension. For instance, in the NAT Transformer block [10], the computation of self attention is $3hwd^2 + 2h^2w^2d$ and that of the FFN layer is $2hwrd^2$, where $r$ is the hidden layer expansion between the two feed-forward layers, $d$ is the token dimension, and the window size is $h \times w$. Thus, the total computation of each NAT Transformer block is as follows:

$$FLOPs(NAT) = 3hwd^2 + 2h^2w^2d + 2hwrd^2. \qquad (3)$$

As shown above, in the total computation of each Transformer block, two terms are quadratic in the token dimension $d$. Since the local attention already reduces the window size ($h$ and $w$), the dimension $d$ becomes the key factor that impacts the total computation.

Based on the above observation, we propose the GVT block design, in which each input token is decomposed into $g$ groups (see Fig. 3). For each such group, its amount of computation (i.e., FLOPs) is calculated as:

$$3hw(d/g)^2 + 2h^2w^2d/g + 2hwr(d/g)^2. \qquad (4)$$

Thus, the total computation of the $g$ groups of a GVT block is:

$$\begin{aligned} FLOPs(GVT) &= g \times (3hw(d/g)^2 + 2h^2w^2d/g + \\ &\qquad 2hwr(d/g)^2) \\ &= \frac{1}{g}(3hwd^2 + 2hwrd^2) + 2h^2w^2d. \end{aligned} \qquad (5)$$

Typically, the computational load attributed to the first two terms in Eq. (4) constitutes the primary portion of the overall computational burden. For example, in the context of the NAT model [10], the combined computation of the first two terms amounts to $4.1 \times 10^6$ FLOPs, while the third term amounts to $4.6 \times 10^5$ FLOPs. Thus, the reduction in computation as described in Eq. (5) significantly alleviates the total computational load.

The group attention structure is shown in Fig. 3. Specifically, for each input feature token $X \in \mathbb{R}^{n \times d}$, we divide the features of $X$ into $g$ groups along its token dimension $d$, resulting in $g$ subsets $X_1, X_2, \ldots, X_g$. On each subset $X_i$, we perform self-attention and FFN separately.

Another advantage of grouping the attention computation is to enhance the diversity of features. As Fig. 3 shows, we split the features of $X$ along the token dimension within each GVT block. During the attention and FFN computation, the features do not communicate across different groups, and thus this process reduces the likelihood of redundancy of features generated by each GVT block.

The details of the GVT block are as follows. Given an input feature token $X \in \mathbb{R}^{n \times d}$, with $n = h \times w$ as the spatial resolution of the feature map, we first split the features of the token into $g$ groups, as:

$$X_i = X[d_i : d_{i+1}], \qquad (6)$$

where $d_i$ and $d_{i+1}$ denote the starting indices of the $i$-th and ($i+1$)-th groups along the token dimension, respectively. The slice operator $[d_i : d_{i+1}]$ returns the segment from index $d_i$ to index $d_{i+1}$, excluding the element at index $d_{i+1}$. Next, we compute attention for each group $i$, as:

$$X_i^{(0)} = Softmax(\frac{Q_i K_i^T}{\sqrt{d_{i+1} - d_i}})V_i. \qquad (7)$$

Afterwards, each $X_i^{(0)}$ is processed by a two-layer feed-forward network, obtaining features $X_i^{(1)}$, as:

$$X_i^{(1)} = \delta(\phi(X_i^{(0)}w_1 + b_1)w_2 + b_2), \qquad (8)$$

where $w_1$ and $w_2$ are the parameters of the two-layer MLP respectively, $\phi$ and $\delta$ are GELU activation functions, and $X_i^{(1)} \in \mathbb{R}^{hw \times (d_{i+1} - d_i)}$. Then, all the features are concatenated and averaged to a 1D vector, as:

$$X^{(2)} = AVG(H(X_1^{(1)}, X_2^{(1)}, \ldots, X_g^{(1)})), \qquad (9)$$

where $H(\cdot)$ denotes concatenation along the token dimension, and $AVG(\cdot)$ denotes average pooling along the $(h, w)$ plane. Thus, $X^{(2)} \in \mathbb{R}^{1 \times d}$.

After the above operation, $X^{(2)}$ is processed by a fully-connected layer, further emphasizing informative groups:

$$s_i = \sigma(X^{(2)} W_3), \qquad (10)$$

where $W_3 \in \mathbb{R}^{d \times (d_{i+1} - d_i)}$ refers to the parameters of the fully-connected layer, $\sigma$ refers to sigmoid activation, and $s_i \in \mathbb{R}^{1 \times (d_{i+1} - d_i)}$. The output of each group $i$ is obtained by rescaling $X_i^{(1)}$, as:

$$X_i^{(3)} = F(X_i^{(1)}, s_i), \qquad (11)$$

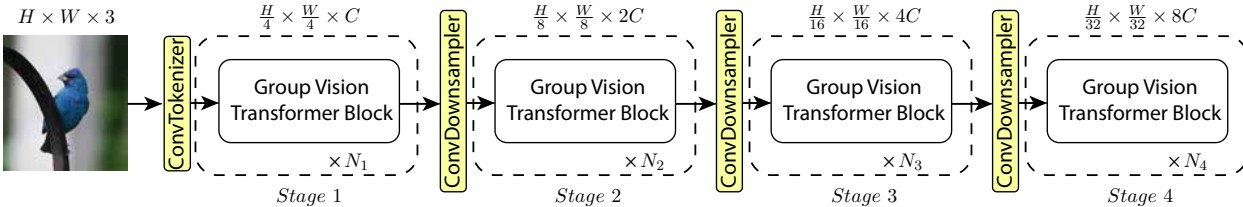

**Figure 2: The overall structure of our proposed Group Vision Transformer model.** $N_1, N_2, N_3,$ **and** $N_4$ **denote the numbers of GVT blocks in the four stages, respectively.** *ConvTokenizer* **is a two-layer strided convolution.** *ConvDownsampler* **is a one-layer strided convolution that increases the channels while reducing the resolution of the feature map.**

**Table 1: Details of different variants of our GVT model. The costs are calculated with an input image size of** $224 \times 224$.

| Variant | # dim | $N_1, N_2, N_3, N_4$ | # heads | # params. | FLOPs |
|---|---|---|---|---|---|
| GVT-Mini | 128 | 3, 4, 6, 5 | 2, 4, 8, 16 | 18.6M | 2.8G |
| GVT-Tiny | 128 | 3, 4, 12, 5 | 2, 4, 8, 16 | 28.1M | 5.2G |
| GVT-Small | 128 | 3, 4, 18, 5 | 2, 4, 8, 16 | 46.3M | 7.5G |
| GVT-Base | 128 | 3, 4, 36, 3 | 2, 4, 8, 16 | 85.9M | 14.5G |

where $F(\cdot)$ is token-wise multiplication between the feature map $X_i^{(1)} \in \mathbb{R}^{hw \times (d_{i+1} - d_i)}$ of group $i$ ($n = h \times w$) and the vector $s_i$. The final output of the GVT block is:

$$X^{(4)} = H(X_1^{(3)}, X_2^{(3)}, \ldots, X_g^{(3)}), \quad (12)$$

The processed feature map $X^{(4)}$ will be sent to the next GVT block for further processing (if there is a next block).

### 3.3 Architecture Variants of GVT

Table 1 presents detailed instantiation information of the model variants of our Group Vision Transformer. Following common practice [10, 12, 21], we construct our base model, denoted as GVT-B, to closely mirror the model parameters and computational complexity of the base model NAT-B in [10]. Also, we build GVT-Mini (GVT-M), GVT-Tiny (GVT-T), and GVT-Small (GVT-S), each tailored to exhibit comparable parameter quantities and computational complexities to their NAT counterparts: NAT-Mini (NAT-M), NAT-Tiny (NAT-T), NAT-Small (NAT-S), and NAT-Base (NAT-B), respectively. We conduct experimental trials to determine optimal configurations for block parameters $N_1, N_2, N_3,$ and $N_4$ (shown in Table 1) to achieve the best performance while retaining the aimed computational complexities. The patch size is set to 4, the token dimension is $d = 128$, the query size is $dim = 128$ (see Table 1), and the MLP expansion ratio is $\alpha = 4$, for all the experiments. The number of groups is set as $g = 4$.

### 4 EXPERIMENTS

We conduct experiments on the ImageNet-1K [7], MS-COCO object detection and instance segmentation [19], and ADE20K semantic segmentation [41] datasets. Our ablation study investigates the effectiveness and computation costs of various network variants of GVT. We employ NAT as the Local Attention block (see Fig. 3).

### 4.1 ImageNet Classification

We train our model variants on the ImageNet-1K dataset to show the effectiveness of our GVT model. ImageNet-1K provides 128K, 50K, and 100K images for training, validation, and testing. The images are categorized into 1000 classes. We use the timm package [33] to train our model. Following the practice of Swin Transformer [21], several data augmentation methods are utilized: CutMix [37], Mixup [38], RandAugment [6], and Random Erasing [40]. We train our model for 300 epochs, with 5 warm-up epochs and an extra 10 cool-down epochs. The initial learning rate of the warm-up stage is $10^{-6}$, the maximum learning rate is $10^{-3}$, and the minimum learning rate is $10^{-5}$. Besides, a cosine learning rate decay strategy is applied. We use AdamW [23] to optimize our model.

Table 2 shows the experimental results comparing our method with several typical methods on the ImageNet-1K dataset: DeiT [30], ConvNeXt [22], PVT [32], Swin [21], NAT [10], BiFormer [42], STViT [15], and CAFormer [36]. The symbols M, T, S, B, and L denote Mini, Tiny, Small, Base, and Large, respectively. Specifically, we use the NAT [10], BiFormer [42], STViT [15], and CAFormer [42] blocks as our base block, and compare with the models without our Group mechanism.

As Table 2 shows, our model demonstrates superior performances compared to the vallina NAT, BiFormer, STViT, and CAFormer, while operating under similar parameter settings and computational constraints. Specifically, our GVT-M, GVT-T, GVT-S, and GVT-B outperform CAFormer-S18, CAFormer-S36, CAFormer-M36, and CAFormer-B36 by 0.4%, 0.3%, 0.2%, and 0.1% in Top-1 Accuracy, respectively. This trend underscores the capability of our Group Vision Transformer to enrich the diversity of the learned features. Further, our GVT block achieves reductions in computations and parameters, enabling our model to preserve more feature maps with similar model parameters and computational costs. Throughout the training process, we partition features into distinct paths with limited inter-path communication. This strategy effectively mitigates feature redundancy, thus enhancing the overall model performance. In addition, we observe a notable enhancement in the Frames Per Second (FPS) of our model compared to its counterpart. This improvement is primarily attributed to splitting the entire Transformer block into multiple groups, each executed in parallel. Consequently, this parallel group processing scheme leads to a significant enhancement in inference speed.

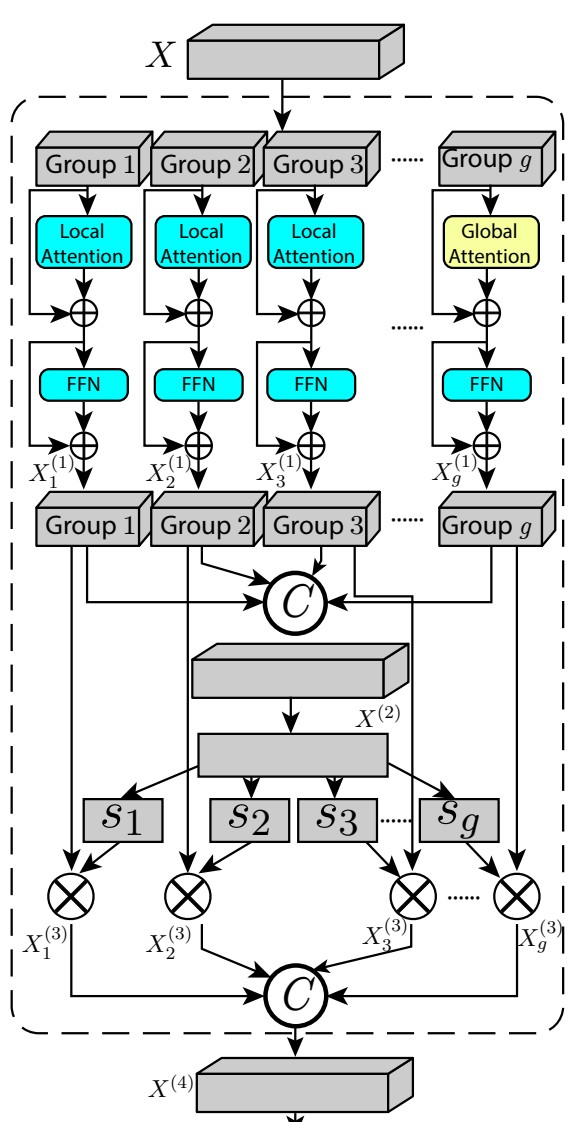

**Figure 3: The structure of our proposed GVT block. $\bigoplus$ denotes summation, $\bigotimes$ denotes token-wise multiplication, and $\copyright$ represents concatenation of feature groups along the token dimension. Local Attention corresponds to the NAT block [10], whereas Global Attention corresponds to the vanilla ViT block [9].**

## 4.2 Object Detection and Instance Segmentation

To further demonstrate the effectiveness of our proposed GVT model, we pre-train our model on the ImageNet-1K dataset and evaluate it on the MS COCO 2017 and ADE20K datasets for object detection, instance segmentation, and semantic segmentation.

We test our pre-trained model on the MS COCO 2017 dataset [19], which contains 118K, 5K, and 20K images for training, validation, and testing, using the Mask R-CNN and Cascade Mask R-CNN frameworks, respectively. Specifically, we pre-train our backbone

**Table 2: Experimental results on the ImageNet-1K dataset, running on 6 NVIDIA RTX 6000 Ada GPUs. The input image size is $224 \times 224$. We use NAT, BiFormer, STViT, and CAFormer as the base local attention, respectively. Bold numbers mark the best performances with different parameters and computations. "*" indicates experimental results acquired using our own GPUs.**

| | Method | # Params. | FLOPs | Thru. (FPS) | Mem. | Top-1 ACC |
|---|---|---|---|---|---|---|
| | DeiT-T [30] | 5.7 M | 1.3 G | 4648 | 64 M | 72.2 |
| | DeiT-S [30] | 22.1 M | 4.6 G | 1723 | 160 M | 79.8 |
| | DeiT-B [30] | 86.6 M | 17.6 G | 535 | 477 M | 81.8 |
| | ConvNeXt-T [22] | 28.0 M | 4.5 G | 1415 | 227 M | 82.1 |
| | ConvNeXt-S [22] | 50.1 M | 8.7 G | 817 | 390 M | 83.1 |
| | ConvNeXt-B [22] | 89.2 M | 15.4 G | 533 | 603 M | 83.8 |
| | PVT-T [32] | 13.2 M | 1.9 G | 1503 | 126 M | 75.1 |
| | PVT-S [32] | 24.5 M | 3.8 G | 964 | 220 M | 79.8 |
| | PVT-L [32] | 61.4 M | 9.8 G | 673 | 476 M | 81.7 |
| | Swin-T [21] | 29.1 M | 4.5 G | 1384 | 223 M | 81.3 |
| | Swin-S [21] | 50.2 M | 8.7 G | 848 | 367 M | 83.0 |
| | Swin-B [21] | 88.1 M | 15.4 G | 621 | 570 M | 83.5 |
| | CSWin-T [8] | 23 M | 4.3 G | 1285 | 209 M | 82.7 |
| | CSWin-S [8] | 35 M | 6.9 G | 1278 | 329 M | 83.6 |
| | CSWin-B [8] | 78 M | 15.0 G | 598 | 604 M | 84.0 |
| | NAT-M [10]* | 20.1 M | 2.7 G | 1708 | 176 M | 81.8 |
| | NAT-T [10]* | 28.0 M | 4.3 G | 1232 | 242 M | 82.9 |
| | NAT-S [10]* | 51.2 M | 7.8 G | 840 | 398 M | 83.6 |
| | NAT-B [10]* | 90.0 M | 13.7 G | 626 | 609 M | 84.1 |
| NAT | GVT-M (ours) | 18.6 M | 2.8 G | 3962 | 229 M | **82.4** |
| | GVT-T (ours) | 28.1 M | 5.2 G | 2725 | 300 M | **83.5** |
| | GVT-S (ours) | 46.3 M | 7.5 G | 1402 | 479 M | **84.1** |
| | GVT-B (ours) | 85.9 M | 14.5 G | 1047 | 836 M | **84.6** |
| | BiFormer-T [42] | 13.1 M | 2.2 G | 1246 | 630 M | 81.4 |
| | BiFormer-S [42] | 26 M | 4.5 G | 647 | 1318 M | 83.8 |
| | BiFormer-B [42] | 57 M | 9.8 G | 419 | 2048 M | 84.3 |
| BiFormer | GVT-T (ours) | 15.5 M | 3.1 G | 2016 | 824 M | **82.6** |
| | GVT-S (ours) | 27.1 M | 5.3 G | 1326 | 1502 M | **84.4** |
| | GVT-B (ours) | 54.5 | 10.9 | 852 | 2446 M | **85.0** |
| | STViT-S [15] | 25 M | 4.4 G | 1096 | 251 M | 83.6 |
| | STViT-B [15] | 52 M | 9.9 G | 555 | 485 M | 84.8 |
| | STViT-L [15] | 95 M | 15.6 G | 373 | 740 M | 85.3 |
| STViT | GVT-T (ours) | 27.1 M | 4.9 G | 2383 | 324 M | **84.0** |
| | GVT-S (ours) | 50.3 M | 10.5 G | 1089 | 520 M | **85.2** |
| | GVT-B (ours) | 96.9 M | 15.8 G | 689 | 872 M | **85.5** |
| | CAFormer-S18 [36] | 26.0 M | 4.1 G | 1638 | 261 M | 83.6 |
| | CAFormer-S36 [36] | 39.0 M | 8.0 G | 890 | 457 M | 84.5 |
| | CAFormer-M36 [36] | 56.0 M | 13.2 G | 667 | 642 M | 85.2 |
| | CAFormer-B36 [36] | 99.0 M | 23.2 G | 504 | 946 M | 85.5 |
| CAFormer | GVT-M (ours) | 27.3 M | 4.0 G | 2830 | 387 M | **84.0** |
| | GVT-T (ours) | 37.9 M | 7.5 G | 1929 | 569 M | **84.8** |
| | GVT-S (ours) | 56.3 M | 14.1 G | 1295 | 834 M | **85.4** |
| | GVT-B (ours) | 100.9 M | 24.0 G | 832 | 1129 M | **85.6** |

model on ImageNet-1K following the practice of NAT [10], and apply the pre-trained backbone to the object detection and instance segmentation frameworks.

We evaluate our method using the backbones of ResNet, ConvNeXt, and Transformers (i.e., Pyramid Vision Transformer (PVT), Swin Transformer (Swin), Neighborhood Attention Transformer (NAT), STViT [15], CAFormer [36], and Cross-Shape Transformer (CSwin)). We utilize the *mmdetection* framework [3] to train our object detection and instance segmentation networks. We resize all the images to have a shorter side of 480 to 800 pixels and a longer side of ≤ 1,333 pixels. Table 3 presents the experimental results on the MS COCO 2017 dataset using the Mask R-CNN framework with the "1×" (12 training epochs) and "3× *LR*" (36 training epochs with multi-scale training) schedules. Table 3 shows that our approach outperforms STViT-B [15] by 1.2% in box AP and 1.1% in mask AP, respectively. On the Tiny model, our approach outperforms STViT-S by 1.2% in box AP and 1.0% in mask AP, respectively.

**Table 3: Experimental results on the MS COCO 2017 dataset using Mask R-CNN [11] with the 1× and 3× training schedules. Images are resized to have a shorter side of 480 to 800 pixels and a longer side of ≤ 1333 pixels. GVT (NAT), GVT (CAFormer), and GVT (STViT) mean that we use NAT, CAFormer, and STViT blocks as the base local attention, respectively. Bold numbers mark the best performances with different parameters and computations. "–" denotes that corresponding results were not reported in the original papers.**

| Method | # Params. (M) | FLOPs (G) | Thru. (FPS) | Mask R-CNN 1× Schedule | | | | | | Mask R-CNN 3× Schedule | | | | | |
|---|---|---|---|---|---|---|---|---|---|---|---|---|---|---|---|
| | | | | $AP^b$ | $AP^b_{50}$ | $AP^b_{75}$ | $AP^m$ | $AP^m_{50}$ | $AP^m_{75}$ | $AP^b$ | $AP^b_{50}$ | $AP^b_{75}$ | $AP^m$ | $AP^m_{50}$ | $AP^m_{75}$ |
| ResNet50 [12] | 44 | 260 | 29 | 38.0 | 58.6 | 41.4 | 34.4 | 55.1 | 36.7 | 41.0 | 61.7 | 44.9 | 37.1 | 58.4 | 40.1 |
| PVT-T [32] | 44 | 245 | 23 | 40.4 | 62.9 | 43.8 | 37.8 | 60.1 | 40.3 | 43.0 | 65.3 | 46.9 | 39.9 | 62.5 | 42.8 |
| Swin-T [21] | 48 | 264 | 21 | 42.2 | 64.6 | 46.2 | 39.1 | 61.6 | 42.0 | 46.0 | 68.2 | 50.2 | 41.6 | 65.1 | 44.8 |
| CSWin-T [8] | 42 | 279 | 19 | 46.7 | 68.6 | 51.3 | 42.2 | 65.6 | 45.4 | 49.0 | 70.7 | 53.7 | 43.6 | 67.9 | 46.6 |
| NAT-T [10] | 48 | 258 | 20 | – | – | – | – | – | – | 47.7 | 69.0 | 52.6 | 42.6 | 66.1 | 45.9 |
| CAFormer-S18 [36] | 45 | 254 | 24 | – | – | – | – | – | – | 48.6 | 70.5 | 53.4 | 43.7 | 67.5 | 47.4 |
| STViT-S [15] | 44 | 252 | 21 | 47.6 | 70.0 | 52.3 | 43.1 | 66.8 | 46.5 | 49.2 | 70.8 | 54.4 | 44.2 | 68.0 | 47.7 |
| GVT-T (NAT) | 40 | 218 | 18 | 47.1 | 69.2 | 51.6 | 43.0 | 66.1 | 45.9 | 49.5 | 70.4 | 54.0 | 44.5 | 68.3 | 46.3 |
| GVT-T (CAFormer) | 47 | 268 | 26 | 47.9 | 70.1 | 52.5 | **43.7** | 67.2 | **47.5** | 50.3 | **71.7** | 54.9 | **45.2** | 69.7 | 47.8 |
| GVT-T (STViT) | 45 | 260 | 26 | **48.1** | 70.6 | 52.9 | 43.7 | 67.5 | 46.9 | **51.4** | 71.4 | **55.6** | 45.2 | 70.2 | 48.1 |
| ResNet100 [12] | 63 | 336 | 22 | 40.4 | 61.1 | 44.2 | 36.4 | 57.7 | 38.8 | 42.8 | 63.2 | 47.1 | 38.5 | 60.1 | 41.3 |
| PVT-S [32] | 64 | 302 | 16 | 42.0 | 64.4 | 45.6 | 39.0 | 61.6 | 42.1 | 44.2 | 66.0 | 48.2 | 40.5 | 63.1 | 43.5 |
| Swin-S [21] | 69 | 354 | 13 | 44.8 | 66.6 | 48.9 | 40.9 | 63.4 | 44.2 | 48.5 | 70.2 | 53.5 | 43.3 | 67.3 | 46.6 |
| ConvNeXt-S [30] | 66 | 262 | 14 | – | – | – | – | – | – | 46.2 | 67.9 | 50.8 | 41.7 | 65.0 | 44.9 |
| CSWin-S [8] | 54 | 342 | 14 | 47.9 | 70.1 | 52.6 | 43.2 | 67.1 | 46.2 | 50.0 | 71.3 | 54.7 | 44.5 | 68.4 | 47.7 |
| NAT-S [10] | 70 | 330 | 15 | – | – | – | – | – | – | 48.4 | 69.8 | 53.2 | 43.2 | 66.9 | 46.5 |
| STViT-B [15] | 70 | 359 | 11 | 49.7 | 71.7 | 54.7 | 44.8 | **68.9** | 48.7 | 51.0 | 72.3 | 56.0 | 45.4 | 69.5 | 49.3 |
| GVT-S (NAT) | 67 | 310 | 18 | 48.2 | 70.3 | 53.1 | 44.0 | 67.5 | 47.0 | 50.3 | 71.8 | 55.3 | 45.0 | 68.3 | 48.2 |
| GVT-S (STViT) | 74 | 371 | 16 | **50.3** | **72.5** | **54.9** | **45.3** | 68.6 | **49.1** | **52.2** | **73.6** | **56.9** | **46.5** | **70.8** | **49.7** |
| ConvNeXt-B [30] | 86 | 741 | 12 | – | – | – | – | – | – | 50.4 | 69.1 | 54.8 | 43.7 | 66.5 | 47.3 |
| PVT-L [32] | 81 | 364 | 14 | 42.9 | 65.0 | 46.6 | 39.5 | 61.9 | 42.5 | 44.5 | 66.0 | 48.3 | 40.7 | 63.4 | 43.7 |
| Swin-B [21] | 107 | 496 | 13 | 46.9 | – | – | 42.3 | – | – | 48.5 | 69.8 | 53.2 | 43.4 | 66.8 | 46.9 |
| CSWin-B [8] | 97 | 526 | 12 | 48.7 | 70.4 | 53.9 | 43.9 | 67.8 | 47.3 | 50.8 | 72.1 | 55.8 | 44.9 | 69.1 | 48.3 |
| STViT-L [36] | 114 | 470 | 10 | 50.8 | 72.5 | 56.3 | 45.5 | 69.7 | 49.1 | 51.7 | 73.0 | 56.9 | 45.9 | 70.4 | 49.9 |
| GVT-B (NAT) | 109 | 512 | 17 | 50.3 | 71.2 | 54.5 | 45.1 | 68.0 | 48.1 | 52.0 | 72.8 | **58.5** | 46.1 | 69.3 | 49.5 |
| GVT-B (STViT) | 120 | 479 | 15 | **51.6** | **73.4** | **57.5** | **46.2** | **71.1** | **50.5** | **52.5** | 73.2 | 57.8 | **47.2** | **71.6** | **50.9** |

**Table 4: Experimental results on the MS COCO 2017 dataset using Cascade Mask R-CNN [2]. Bold numbers mark the best performances with different parameters and computations.**

| Method | # Params. | FLOPs | Cascade Mask R-CNN 3× schedule | | | | | |
|---|---|---|---|---|---|---|---|---|
| | | | $AP^b$ | $AP^b_{50}$ | $AP^b_{75}$ | $AP^m$ | $AP^m_{50}$ | $AP^m_{75}$ |
| NAT-M [10] | 77 | 704 | 50.3 | 68.9 | **54.9** | 43.6 | 66.4 | 47.2 |
| GVT-M (NAT) | 68 | 701 | **52.0** | **72.2** | 54.2 | **44.5** | **66.8** | **48.4** |
| ResNet50 [12] | 82 | 739 | 46.3 | 64.3 | 50.5 | 40.1 | 61.7 | 43.4 |
| ResNext101 [35] | 101 | 819 | 48.1 | 66.5 | 52.4 | 41.6 | 63.9 | 45.2 |
| Swin-T [21] | 86 | 745 | 50.5 | 69.3 | 54.9 | 43.7 | 66.6 | 47.1 |
| ConvNeXt-T [30] | 86 | 741 | 50.4 | 69.1 | 54.8 | 43.7 | 66.5 | 47.3 |
| CSWin-T [8] | 80 | 757 | 52.5 | 71.5 | 57.1 | 45.3 | 68.8 | 48.9 |
| NAT-T [10] | 85 | 737 | 51.4 | 70.0 | 55.9 | 44.5 | 67.6 | 47.9 |
| CAFormer-S18 [36] | 81 | 733 | 52.3 | 71.3 | 56.9 | 45.2 | 68.6 | 48.8 |
| STViT-B [15] | 108 | 837 | 53.9 | 72.7 | **58.5** | 46.8 | **70.4** | 50.8 |
| GVT-T (NAT) | 80 | 720 | 52.3 | 71.5 | 57.0 | 45.1 | 68.4 | 49.5 |
| GVT-S (CAFormer) | 78 | 714 | 53.0 | 72.5 | 58.0 | 45.9 | 69.4 | 50.2 |
| GVT-S (STViT) | 116 | 853 | **54.3** | **73.4** | 58.2 | **47.1** | 70.1 | **51.2** |
| ConvNeXt-B [30] | 146 | 964 | 52.7 | 71.3 | 57.2 | 45.6 | 68.9 | 49.5 |
| Swin-B [21] | 145 | 982 | 51.9 | 70.5 | 56.4 | 45.0 | 68.1 | 48.9 |
| NAT-B [10] | 147 | 931 | 52.5 | 71.1 | 57.1 | 45.2 | 68.6 | 49.0 |
| CAFormer-M36 [36] | 132 | 920 | 53.8 | 72.5 | 58.3 | 46.5 | 70.1 | 50.7 |
| GVT-B (NAT) | 140 | 922 | 53.4 | 72.1 | 57.9 | 46.4 | 69.6 | 50.1 |
| GVT-B (CAFormer) | 134 | 928 | **54.6** | **73.7** | **59.1** | **47.6** | **70.9** | **51.5** |

Table 4 reports the object detection and instance segmentation results on the MS COCO 2017 dataset using the Cascade Mask R-CNN [2] framework. Likewise, we find that our GVT model achieves promising performance gains compared to the known methods. Specifically, our GVT-B (CAFormer) and GVT-S (CAFormer) outperform CAFormer-M36 and CAFormer-S18 by 0.8% and 0.7% in

box AP, respectively. The mask mAP performances also demonstrate the superiority of our model over the known ones. Since our GVT model and the NAT model use the same Local Attention block, these performance gains manifest the effectiveness and consistent improvement of our GVT approach.

## 4.3 Semantic Segmentation

To demonstrate the generalizability of our GVT on semantic segmentation, we train our model using UperNet [34] on the ADE20K dataset. Specifically, we pre-train our GVT on the ImageNet-1K dataset, and apply the pre-trained backbone to UperNet for semantic segmentation. For fair comparison, we follow the practice of Swin Transformer and NAT by using the *mmsegmentation* framework [5] for training and testing. We train all the models for 160K iterations and set the batch size to 16. The training images are randomly resized and cropped to 512 × 512 in the training stage. During the test stage, a multi-scale testing using resolutions which are [0.5, 0.75, 1.0, 1.25, 1.5, 1.75] multiplied with that used in the training stage is employed.

Table 5 presents the experimental results on ADE20K semantic segmentation in the mIoU measure. We compare our method with ConvNeXt [22], Swin Transformer [21], CSWin Transformer [8],

**Table 5: Experimental results on the ADE20K dataset of semantic segmentation using UperNet [34]. Bold numbers mark the best performances with different parameters and computations. Training images are randomly resized and cropped to size 512×512. In the test stage, a multi-scale testing using resolutions which are [0.5, 0.75, 1.0, 1.25, 1.5, 1.75] multiplied with that used in the training stage is employed.**

| Method | # Params. (M) | FLOPs (G) | Thru. | mIoU |
|---|---|---|---|---|
| NAT-M [10] | 50 | 900 | 25 | 46.4 |
| GVT-M (ours) | 48 | 895 | 24 | **47.2** |
| Swin-T [21] | 60 | 946 | 21 | 45.8 |
| ConvNeXt-T [30] | 60 | 939 | 21 | 46.7 |
| CSWin-T [8] | 60 | 959 | 19 | 49.3 |
| NAT-T [10] | 58 | 934 | 20 | 48.4 |
| SViT-S [15] | 54 | 926 | 20 | 48.6 |
| CAFormer-S18 [36] | 54 | 1024 | 21 | 48.9 |
| GVT-T (NAT) | 58 | 930 | 21 | 49.9 |
| GVT-T (CAFormer) | 55 | 1036 | 19 | **50.1** |
| Swin-S [21] | 81 | 1040 | 17 | 49.5 |
| ConvNeXt-S [30] | 82 | 1027 | 17 | 49.6 |
| NAT-S [10] | 82 | 1010 | 15 | 49.5 |
| SViT-B [15] | 80 | 1036 | 15 | 50.7 |
| CAFormer-S36 [36] | 67 | 1197 | 11 | 50.8 |
| GVT-S (NAT) | 80 | 1027 | 14 | 50.4 |
| GVT-S (CAFormer) | 69 | 1228 | 11 | **52.0** |
| Swin-B [21] | 121 | 1188 | 15 | 49.7 |
| ConvNeXt-B [30] | 122 | 1170 | 14 | 49.9 |
| CSWin-B [8] | 109 | 1222 | 13 | 51.1 |
| NAT-B [10] | 123 | 1137 | 14 | 49.7 |
| STViT-L [15] | 125 | 1151 | 14 | 52.4 |
| CAFormer-M36 [36] | 84 | 1346 | 10 | 51.7 |
| GVT-B (NAT) | 105 | 1210 | 14 | 52.4 |
| GVT-B (CAFormer) | 86 | 1401 | 9 | **52.8** |

and NAT [10]. As Table 5 shows, our GVT outperforms the state-of-the-art models in different variants. Specifically, our GVT-T, GVT-S, and GVT-B achieve 1.2%, 1.2%, and 1.1% performance gains over the counterparts of CAFormer [42] with similar parameters and computational complexities.

## 4.4 Feature Map Diversity

To further demonstrate the capability of our method in diversifying feature maps, we compare the dissimilarity [1] of the generated feature maps by our model and the vanilla base models (NAT-B [10], CAFormer-M36 [36], BiFormer-B [42], and STViT-L [15]). We use the cosine distance to measure the dissimilarity between two features $f_i$ and $f_j$ in a feature map $f \in \mathbb{R}^{C \times H \times W}$, with $f_i, f_j \in R^{H \times W}$. The diversity of $f$ is calculated as:

$$Dis(f) = \frac{1}{m} \sum_{i=1}^{C} \sum_{j=1}^{i-1} cos(f_i, f_j), \quad (13)$$

$$m = \frac{C*(C-1)}{2}, \quad (14)$$

**Table 6: Feature map dissimilarity between our method and previous methods. GVT (NAT), GVT (CAFormer), GVT (Bi-Former), and GVT (STViT) denote using NAT, CAFormer, BiFormer, and STViT blocks as the local attention, respectively.**

| Method | Dissimilarity |
|---|---|
| NAT-B [10] | 0.364 |
| GVT (NAT) | 0.487 |
| CAFormer-M36 [36] | 0.649 |
| GVT (CAFormer) | 0.798 |
| BiFormer-B [42] | 0.761 |
| GVT (BiFormer) | 0.884 |
| STViT-L [15] | 0.774 |
| GVT (STViT) | 0.896 |

where $m$ is the number of feature pairs, and $cos(f_i, f_j)$ is the cosine distance between two features $f_i$ and $f_j$ in $f$.

The feature map dissimilarities of NAT-B, CAFormer-M36, BiFormer-B, STViT-L, and our models are shown in Table 6. Observably, our Group mechanism consistently enhances the diversity of the extracted feature maps due to the fact that the Group mechanism reduces the likelihood of redundancy of features generated by each GVT block. A qualitative example of feature map comparison between CAFormer-M36 and our GVT with the CAFromer block as the base local attention is shown in Fig 4.

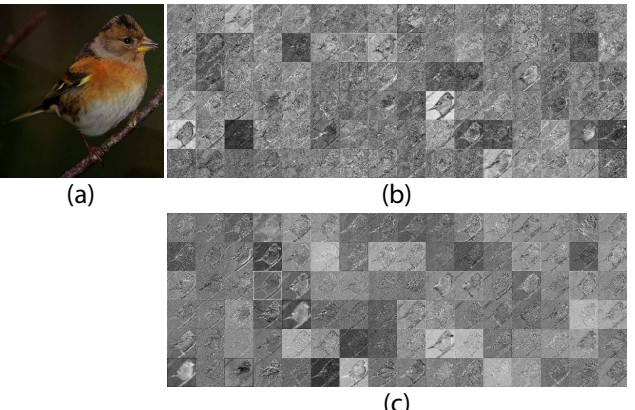

**Figure 4: An example of feature map comparison between our method and CAFormer-M36. (a) An input image; (b) the feature map generated by CAFormer-M36; (c) the feature map generated by our GVT-S (CAFormer) model.**

## 4.5 Ablation Study

In this section, we conduct ablation study on the proposed GVT approach. Specifically, we explore how the Self Attention (SA) Grouping, FFN Grouping, and Re-scaling module affect the model performance respectively on the ImageNet-1K dataset. The results are shown in Table 7. Note that we adjust the number of channels of each stage to closely mirror the model parameters and computational complexity of CAFormer-S18. Observably, combining the

**Table 7: Ablation study on the effects of different key components of our proposed GVT approach. The CAFormer [36] block is utilized as the local attention.**

| Setting | ImageNet-1K | | | MS COCO | | ADE-20K |
|---|---|---|---|---|---|---|
| | # Params | FLOPs | Top-1 ACC | $AP^b$ | $AP^m$ | mIoU |
| CAFormer-S18 | 26.0 | 4.1 | 83.6 | 48.6 | 43.7 | 48.9 |
| w/o SA Grouping | 28.4 | 4.2 | 83.8 | 49.5 | 45.1 | 48.9 |
| w/o FFN Grouping | 29.2 | 4.5 | 83.7 | 49.1 | 44.8 | 48.6 |
| w/o Re-scaling | 28.2 | 4.6 | 83.8 | 49.4 | 45.0 | 49.3 |
| GVT-T (CAFormer) | 27.3 | 4.0 | 84.0 | 50.3 | 45.2 | 50.2 |

**Table 8: Ablation study of the efficacy of the number of local attention groups. We designate the total number of groups to be 8.**

| | # of Local Attention Groups | | | | | | | |
|---|---|---|---|---|---|---|---|---|
| | 1 | 2 | 3 | 4 | 5 | 6 | 7 | 8 |
| # Params (M) | 59.4 | 58.9 | 58.3 | 57.6 | 56.8 | 56.3 | 56.2 | 56.1 |
| FLOPs (G) | 17.9 | 17.1 | 16.4 | 15.8 | 15.0 | 14.1 | 13.9 | 13.6 |
| Top-1 ACC | 82.7 | 83.3 | 83.5 | 83.9 | 84.8 | 85.4 | 85.2 | 84.9 |

SA Grouping, FFN Grouping, and Re-scaling gives rise to the best performance.

We further examine the efficacy of global and local grouping. Employing CAFormer-M36 as the local attention model, we designate the total number of groups to be 8. The results are shown in Table 8. Notably, a performance enhancement is observed as the number of local attention groups initially increases. Furthermore, the performance saturates when the number of local attention groups reaches 6, thereafter experiencing a decline as the number of local attention groups continues to increase. This can be attributed to the complementary roles played by both local attention, which captures fine-grained details, and global attention, which encapsulates broader contextual information, thereby contributing to the overall image representation capability [24].

For fair comparison, we integrate our GVT mechanism into the backbone architectures of both Swin Transformer [21] and NAT [10], respectively. Our evaluations are conducted on the ImageNet-1K dataset.

Compared to Table 8 that studies the effect of the ratio of Local Attention Groups, in Table 9, we present the effects of the total number of groups and the token dimension of each group, denoted as "GVT_$N \times W$", where $N$ denotes the number of groups and $W$ denotes the token dimension of each group. Note that for an input feature token $X$, the token dimension $d$ of $X$ is $d = N \times W$ (for all the $N$ groups). As Table 9 shows, more groups give rise to performance gains under similar parameter numbers and computational budgets. On the backbone of Swin Transformer, our GVT ($N = 4, W = 32$) outperforms its counterparts by 1.3%, 0.5%, and 0.4% in Top-1 Accuracy, respectively. On the backbone of Neighborhood Attention Transformer (NAT), our GVT ($N = 4, W = 32$) outperforms its counterparts by 0.6%, 0.6%, 0.5%, and 0.5% in Top-1 Accuracy, respectively. These experimental results demonstrate the performance effectiveness and computational advantages of our approach.

Notably, performance enhancements are observed with increased group numbers, attributing to the capability of our Group Vision

**Table 9: Ablation study of utilizing two different backbones on the ImageNet-1K dataset. (Swin) and (NAT) denote using the Swin Transformer and NAT blocks as the Local Attention, respectively. Bold numbers mark the best performances with different parameters and computations.**

| Method | # Params. (M) | FLOPs (G) | Top-1 ACC |
|---|---|---|---|
| Swin-T [21] | 29.1 | 4.5 | 81.3 |
| GVT-T_2 × 48 (Swin) | 28.3 | 4.6 | 82.4 |
| GVT-T_4 × 32 (Swin) | 28.1 | 4.3 | **82.6** |
| GVT-T_8 × 24 (Swin) | 28.5 | 4.8 | **82.6** |
| Swin-S [21] | 50.2 | 8.7 | 83.0 |
| GVT-S_2 × 48 (Swin) | 50.4 | 8.5 | 83.3 |
| GVT-S_4 × 32 (Swin) | 49.8 | 8.7 | 83.5 |
| GVT-S_8 × 24 (Swin) | 50.1 | 8.5 | **83.6** |
| Swin-B [21] | 88.1 | 15.4 | 83.5 |
| GVT-B_2 × 48 (Swin) | 87.5 | 14.6 | 83.6 |
| GVT-B_4 × 32 (Swin) | 87.9 | 15.0 | 83.9 |
| GVT-B_8 × 24 (Swin) | 88.5 | 15.5 | **84.0** |
| NAT-M [10] | 20.1 | 2.7 | 81.8 |
| GVT-M_2 × 48 (NAT) | 19.5 | 3.2 | 82.0 |
| GVT-M_4 × 32 (NAT) | 18.6 | 2.8 | 82.4 |
| GVT-M_8 × 24 (NAT) | 19.2 | 3.5 | **82.5** |
| NAT-T [10] | 28.0 | 4.3 | 82.9 |
| GVT-T_2 × 48 (NAT) | 29.2 | 4.8 | 83.1 |
| GVT-T_4 × 32 (NAT) | 28.1 | 5.2 | 83.5 |
| GVT-T_8 × 24 (NAT) | 28.5 | 5.5 | **83.6** |
| NAT-S [10] | 51.2 | 7.8 | 83.6 |
| GVT-S_2 × 48 (NAT) | 48.5 | 8.0 | 83.9 |
| GVT-S_4 × 32 (NAT) | 46.3 | 7.5 | 84.1 |
| GVT-S_8 × 24 (NAT) | 48.0 | 8.2 | **84.2** |
| NAT-B [10] | 90.0 | 13.7 | 84.1 |
| GVT-B_2 × 48 (NAT) | 87.2 | 15.5 | 84.3 |
| GVT-B_4 × 32 (NAT) | 85.9 | 14.5 | **84.6** |
| GVT-B_8 × 24 (NAT) | 88.0 | 16.7 | **84.6** |

Transformer to enrich the learned feature representations. Furthermore, our model exhibits more reductions in parameters and computations when more groups are used. Consequently, under comparable parameters and computational budgets, our model can accommodate a larger token dimension, thereby generating more feature maps and yielding superior performance.

## 5 CONCLUSIONS

In this paper, we introduced the *Group Vision Transformer* (GVT), a novel and efficient fundamental framework. By conducting self-attention and utilizing feed-forward network on distinct groups of feature tokens, we reduced computational complexity and model parameters. Concurrently, the utilization of multi-group self-attention fosters enhanced feature diversity and reduces feature redundancy. Experiments on several computer vision tasks, including ImageNet-1K classification, MS COCO 2017 object detection and instance segmentation, and ADE20K semantic segmentation, validated the efficacy of our proposed GVT model. Ablation study conducted on these tasks and various backbones verified the effectiveness of the proposed GVT mechanism.

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
