# OpenReview forum: "Group Vision Transformer"
_acmmm.org/ACMMM/2024/Conference — MM2024 Poster_

### Official Review · Reviewer_s5PK · 2024-05-14

**Rating:** 4
**Confidence:** 3

**Summary:**

The paper points out the importance of reducing the computation demands of the Feedforward layer, which most researchers have overlooked. They introduce the Group Vision Transformer (GVT)  block to split the features into G groups, and further, each group is processed independently during the Attention and Feedforward layer. By doing this, the computation complexity is significantly reduced and also achieves comparable performance. This work emphasizes the importance of reducing the compute complexity of the Feedforward layer and producing more diverse features which may contribute to better performance. Extensive experiments demonstrate their effectiveness.

**Strengths:**

1.	The paper introduces a method reducing parameters and computational complexity by splitting token embedding into different groups and computing them independently during the subsequent blocks.

2.	The proposed model, GVT, outperforms existing models on several benchmarks, providing solid empirical evidence for the effectiveness of their approach.

3.	They found that the independent branches design can produce more diverse feature maps and benefit downstream tasks.

**Limitations:**

1.	The proposed method found that independent branches produce more diverse features. They lack experiments or theoretical guarantees to justify whether total independent or partial dependent groups are better. Will different group communication strategies further improve the model performance?

2.	For group splits, the authors should further clarify the influence of different group numbers and the correlation between group numbers and other factors such as model size and dataset size.

3.	There is a lack of analysis of different group sizes that influence attention mechanisms and a lack of ablation study of the global attention branch. Will a simple non-overlap split cause information loss for the following attention blocks?

**Suitability:**

2

---

### Official Review · Reviewer_sppK · 2024-05-23

**Rating:** 3
**Confidence:** 3

**Summary:**

This paper introduces a new visual backbone network called the Group Vision Transformer (GVT). The primary idea is to independently divide the Vision Transformer into multiple groups and then perform self-attention and feed-forward network (FFN) operations independently within each group.

**Strengths:**

1. Through grouped operations, the Group Vision Transformer (GVT) can significantly reduce the number of parameters and GFLOPS of the original Vision Transformer. In the design of model frameworks with equal numbers of parameters and GFLOPS, GVT surpasses other visual backbone networks in various visual tasks.

2. The method proposed in the paper is simple and effective, with sufficiently comprehensive experiments.

**Limitations:**

1. In the Transformer, there has been some prior work on mimicking Group Convolution to perform grouped operations on transformer blocks, such as [1] and [2]. The authors should compare and analyze their work with these papers.

2. With the same number of parameters and GLOPS as other models, the introduction of the Group operation enables GVT to add larger feature dimensions and design deeper network. However, this approach results in a substantial increase in GPU memory usage. For instance, as shown in Table 2, the memory consumption of GVT-M is 48% higher compared to CAFormer-S18, and the memory usage for other model sizes has increased by more than 20%. This comparison seems to be somewhat unfair.

[1] Park S, Kim G, Lee J, et al. Scale down Transformer by Grouping Features for a Lightweight Character-level Language Model[C]//Proceedings of the 28th International Conference on Computational Linguistics. 2020: 6883-6893.

[2] Liu X, Peng H, Zheng N, et al. Efficientvit: Memory efficient vision transformer with cascaded group attention[C]//Proceedings of the IEEE/CVF Conference on Computer Vision and Pattern Recognition. 2023: 14420-14430.

**Suitability:**

2

---

### Official Review · Reviewer_AYns · 2024-05-25

**Rating:** 5
**Confidence:** 3

**Summary:**

The authors conducted self-attention and utilized a feed-forward network on distinct groups of feature tokens for reducing computational complexity, and model parameters. The utilization of multi-group self-attention fosters enhanced feature diversity and reduces feature redundancy. Experiments on several computer vision tasks, including ImageNet-1K classification, MS COCO 2017 object detection and instance segmentation, and ADE20K semantic segmentation, validated the efficiency of the proposed GVT model. The core idea of this model aims to divide a global transformer operation into multiple independent branches that effectively disentangle information entanglement. And, it is more useful for vision tasks due to its part-whole relations.

**Strengths:**

The model demonstrates superior performances compared to the NAT, BiFormer, STViT, and CAFormer while operating under similar parameter settings and computational constraints.  This strategy effectively mitigates feature redundancy, thus enhancing the overall model performance.

**Limitations:**

This kind of idea has been a lot in the CNN era. I want to know what the author thinks about transferring the idea from the CNN era to the transformer. What is the difference or what are the advantages and disadvantages?

**Suitability:**

2

---

### Meta-Review · Area_Chair_2WME · 2024-06-28

**Recommendation:** Accept (Poster)
**Confidence:** 5

**Metareview:**

This paper introduced a simple and efficient variant of Vision Transformer that divides and groups the entire Transformer layer to improve attention computation, showing competitive performances in common vision tasks and outperforming CAFormer-S36 with similar parameters and complexity. The reviewers provided positive feedback on the paper's novelty, methodology, and results, with some suggestions for further analysis and clarification. Overall, the paper is technically solid and will have high impact on multimodal perception research. Given the resolution of raised concerns and the unanimous positive reviews, the paper is accepted for publication.